# Will the Construction of Sports Facilities Nudge People to Participate in Physical Exercises in China? The Moderating Role of Mental Health

**DOI:** 10.3390/healthcare11020219

**Published:** 2023-01-11

**Authors:** Xiaojing Xue, Yong Li

**Affiliations:** 1School of Public Administration, Zhejiang University of Technology, No. 288 Liuhe Road, Xihu District, Hangzhou 310023, China; 2School of Marxism, Shanghai Maritime University, No. 1550 Haigang Avenue, Pudong New Area, Shanghai 201306, China

**Keywords:** sports facilities, physical exercises, mental health, nudge, moderating effect

## Abstract

This study aims to examine the nudging effect of the sports facility construction on physical exercise (PE) participation with consideration of the moderating role of mental health in China. Multiple linear regression models are used in this study. The subjects are 4634 from the 2014 China Family Panel Studies (CFPS) data, which is a nationally representative longitudinal survey of Chinese individuals. We find that the construction of sports facilities nudges people to participate in PE, and gender, age, and education significantly influence people’s participation in PE. Young, female, and better-educated people compose the “neo-vulnerable” population, who participate less in PE in China and need more interventions. Mental health status has no significant effect on people’s PE participation, while it negatively moderates the nudging effect of the construction of sports facilities on PE. The results of this study suggest that only building sporting facilities is insufficient to encourage PE participation. Policies and interventions should be given to mentally disturbed individuals to guarantee and magnify the nudging effect of sports facilities on PE.

## 1. Introduction

The World Health Organization (WHO) advocates physical activity and exercise as one of the best approaches to reducing health risks and promoting healthy lifestyles [1]. However, the overall progress is slow [2]. WHO highlights that almost 500 million people will develop heart disease, obesity, and diabetes caused by physical inactivity between 2020 and 2030, costing USD 27 billion annually, if governments do not take urgent action to encourage more PE in their countries [2]. A survey of Norway shows that a large majority of young adults fail to meet international recommendations on PE, posing a significant challenge to public health [3]. Countries are encouraged to design and implement policies and measures to raise levels of PE [2]. Because of the COVID-19 pandemic, the trend of physical inactivity is supposed to continue for several years [4]. Athletes’ training and competition at all levels have been suspended, and fitness centers in many countries have closed [5]. Research from Spain find that PE decreases significantly during COVID-19 confinement [6]. It is also found that the COVID-19 pandemic has been associated with a substantial rise in symptoms of mental disorders, such as anxiety, and worries, which lower people’s interest in PE participation [7,8].

Due to rising urbanization in recent years, China faces rising levels of obesity and physical inactivity [9]. Thus, how to promote PE, including mass and elite sports in China, receives increasing attention [9]. For example, young elite athletes in China sacrifice their education of other skills to win glory for the country and experience the psychological issues of medal-orientation pressure and career transition crises because of the lack of full-fledged dual career programs [10]. Since the establishment of the People’s Republic of China, “whole country support for the elite sport system” was set up to pursue elite sports success [11]. Under this system, young talented athletes are developed in a relatively closed system, which is separate from the education system. In the future, academic sports should ensure that young elite athletes are equipped with basic employability skills that can be adapted to future work-related conditions and challenges [12].

Nudging people to participate in PE by setting up a feasible environment is one of the core pillars of Chinese sports policies, which is in line with the health promotion strategy advocated by WHO [13]. Similar policies have been applied in many countries and regions worldwide, such as Health Promoting Health Service (HPHS) in Scotland [14] and healthy city interventions in Colombia and Chile [15]. A qualitative study conducted in Canada suggests that a built environment within neighborhoods and cities is essential for maintaining people’s health via encouraging outdoor PE [16]. Closer distances between an individual’s home and sports facilities are associated with high levels of physical activities [17]. A growing number of studies report that the accessibility of sports facilities can positively affect individuals’ participation in PE, such as in Korea, Norway, Sweden, and New England [18,19,20,21]. What the relationship is between sports facility construction and PE participation in China remains to be answered.

Mental health is another major problem in China. According to Report on National Mental Health Development in China (2019–2020), the heavy psychological pressure is the most distinct aspect of the national health status in 2021 [22]. General people are bothered by depression, anxiety, and stress from family and work, although they are not diagnosed as mental illness occupants. These bad moods are considered as barriers to physical activities [23]. Will the construction of sports facilities, especially settled in residential areas, encourage people to participate in physical activities under the heavy pressure of psychological disorders, such as subclinical depression and anxiety, in China? How will mental health status moderate the nudging effect of sports facilities construction on people’s PE participation?

## 2. Literature Review

Theoretically, this study is based on the health promotion model (HPM) proposed by Pender [24]. HPM indicates that both situational and personal factors can contribute to individuals’ PE participation [25].

### 2.1. Construction of Sports Facilities and PE Participation

Situational factors include the options that are perceived as being available environmental features; for example, when the helmet is available, one can opt to ride a motorcycle if he/she wants [26]. Built environments could be nudge architectures to induce positive behavioral change [27]. Environments designed for physical activities, such as sports facilities, could have a positive influence on individual choice of physical activities [28]. People with easier access to sport facilities are reported to have a stronger tendency to participate in PE [29]. Having parks, green spaces, or exercise facilities close to residence may lead to an increasing level of recreational PE [30]. As the living place for general people, the neighborhood environment is emphasized in physical exercise promotion [31]; for instance, a study in Canada shows that neighborhood characteristics are strongly related to physical exercise outcomes [32]. A study conducted in Brazil also suggests recreational facilities or venues along the route (e.g., parks) are contributive factors associated with youth PE [33]. A study from Mexico suggests that environments with better lighting could increase physical activity in female youth [34]. Well-connected street networks with secure pedestrian and cycling infrastructure may improve neighborhood walkability [35]. On the other hand, low accessibility of sports facilities is regarded as a barrier toPE. It is found that individuals in rural areas are less likely to reach the USA-government-recommended levels of PE due to a lack of sports facilities [36]. People living in disadvantaged areas may have poor tax bases with which to finance recreation sports facilities that enable individuals to participate in PE [37,38]. Therefore, we propose:

**Hypothesis** **1** **(H1).**
*Construction of sports facilities positively affects people’s PE participation.*


### 2.2. Mental Health and PE Participation

People’s mental health status is regarded to be positively related to PE participation [39]. Mental health status is considered as either a barrier or a motivator to participating in physical activities. People with mental illness may encounter difficulties advancing their involvement in PE. [40,41,42]. A study conducted in the USA suggests that poorer mental health, such as COVID-19-related worry and stress, is associated with lower PE [43]. Meanwhile, it is found that people with mental illness enjoy PE, believe in its benefits, and have a desire to be more active in physical activities [44]. Included as a part of mental health treatment, PE is regarded as a valuable therapeutic component [45]. To be specific, perceived benefits of PE include improved sleep, feelings of accomplishment, relaxation, higher energy levels, and improved self-confidence [46,47,48]. In light of the above divergent findings, the following hypotheses are proposed:

**Hypothesis** **2a** **(H2a).**
*Mental health negatively affects people’s PE participation.*


**Hypothesis** **2b** **(H2b).**
*Mental health positively affects people’s PE participation.*


### 2.3. The Moderating Effect of Mental Health

The effect of mental health on health activities could be indirect [49,50]. The poor mental health may be associated with reduced motivation in PE participation, even with sufficient sports facilities. Perception of unsafe traffic and crime safety may result in more time spent indoors [51]. Although the relationship between the construction of sports facilities and PE participation has received increasing attention, most studies have been conducted in Western or developed countries [32,40]. The situations in developing countries in Asia, especially in China, are still left for empirical discussions. In addition, the existing literature emphasizes the inactive trend of people with severe mental illness, while rarely being about general people suffering from psychological disturbance [52]. This population needs more academic attention. Thus, in this study, mental health status is used as a moderator and the following hypothesis is proposed:

**Hypothesis** **3** **(H3).**
*The relationship between construction of sports facilities and people’s PE participation varies by their mental health status.*


## 3. Data and Methods

The purpose of this study is to analyze the function of the construction of sports facilities on PE participation and the moderating role of mental health status. This study analyzes data from CFPS, which is a nationally representative survey of Chinese individuals launched in 2008 by the Institute of Social Science Survey of Peking University, China [53]. The CFPS is designed to collect longitudinal data in contemporary China. This survey focuses on the economic, as well as the non-economic, wellbeing of the Chinese population, with a wealth of information covering such topics as economic activities, education outcomes, family dynamics and relationships, migration, and health [54]. The sampling method of CFPS is implicit stratification [54]. To ensure the rights and interests of the interviewees participating in the project, CFPS complies with the provisions of Biomedical Ethics Committee of Peking University [55]. The ethical review batch number of CFPS project is IRB00001052-14010 [55]. The CFPS received consent from study participants before the beginning of the study. The data used in this study are 2014 CFPS, which include information from a total of 29,656 people aged 16 or above. Individuals with data missing for density of sports facilities, physical activity, depression, BMI, and marital status and education are excluded. The final sample population of 4634 individuals was selected for this study.

As for the dependent variable, the response to the question in CFPS survey “how much time do you spend doing PE weekly (hour)?” is used to evaluate the level of PE participation. PE here includes walking, long-distance running, jogging, mountain climbing, practicing martial arts such as Taijiquan and Qigong, indoor dance, dance aerobics, yoga, ball games, swimming, diving, rowing, sailing a boat and other water sports, winter ice and snow sports, wrestling, judo, boxing, and other physical contact sports [56]. Regarding the independent variable, the construction of sport facilities is evaluated by density of sports facilities in the community, similar to the measurement adopted in a previous study [57]. The referred questions in CFPS survey are “how many sport facilities are there in your village/residential community?” and “What is the current administrative area of your village/residential in square kilometers?” The density of sports facilities equals the number of sports facilities divided by the area of the community.

The mental health status is aggregated from the factors of depression, meaningless, hopeless, and fatigue. These factors are evaluated, respectively, by the response to the question “how often during the past month did you feel depressed/meaningless/hopeless?” and “how often during the past month did you feel that everything was an effort?” The Cronbach’s α of mental health status is 0.78, indicating acceptable internal consistency [58,59].

This study performs OLS regression analysis to identify the relationships between the construction of sports facilities, mental health status, and PE by controlling potential confounders, including age, gender, education, marital status, self-rated health, BMI. The statistical analysis software used in this study is Stata version 17.

## 4. Results

Table 1 shows the summary of the sample. Gender is coded into a binary variable: 1 = “male” and 0 = “female”. Age is divided into three groups: 1 = “16–40”, 2 = “40–65”, and 3 = “65–90”. As demonstrated in Table 1, the average time of respondents’ PE is 7.621 h weekly. The mean density of sports facilities is 1.899, indicating a lack of built sports environment. The average mental health status of the investigated people is 4.55.

Table 2 presents the Pearson correlation matrix of this study. The correlations among PE participation, construction of sports facilities, and mental health status are all significant.

Table 3 shows the results of OLS regression analysis. As shown in Model 4 of Table 3, the construction of sports facilities has a significantly positive effect on people’s PE participation (β = 1.448, *p* < 0.05). H1 is supported. People resident in the environment equipped with a high density of sport facilities participate more in PE.

The mental health status is found to have no significant relationship with PE participation (β = 0.219, *p* > 0.05), while its moderating effect exists (β = −0.275, *p* < 0.05). Therefore, H2a and H2b are not supported, and H3 is verified. Mental ill-health restrains the nudging effect of sports facilities. In the case of low level of mental health, the contribution of sports facilities to PE participation is weakened. People suffering from psychological disturbance will be limitedly benefited by the construction of sports facilities.

Gender is significantly positively related to PE participation (β = 0.679, *p* < 0.05) (see Model 4). Men participate more in PE compared with women. As shown in Model 4, age also has a significantly positive effect on PE participation (β = 0.133, *p* < 0.001). Older people are more likely to participate in PE. One possible explanation for this finding is that older people have more leisure time after retirement.

Meanwhile, educational level has a negative effect on the PE participation (β = −0.214, *p* < 0.001) (see Model 4). People with higher educational level tend to perform less PE in China. One possible explanation for this finding may be that well-educated people are busier and have less leisure time for PE participation.

The self-rated health is found to have no significant effect on PE participation (β = −0.177, *p* > 0.05). Neither does BMI (β = 0.005, *p* > 0.05) in this study. People’s behavior of PE may barely have a relationship with their health status. Those who are in good health status may feel no incentives from their bodies to perform PE and those who have poor health may have the intention but no capability to participate in PE. The relationship between self-rated health status and PE participation needs more detailed research in the future.

## 5. Discussion

The results of this study suggest that the construction of sports facilities has a nudging effect on people’s PE participation, achieving the objectives of this study and confirming the HPM theory with China’s empirical study [25]. The construction of sports facilities in neighborhoods is useful to encourage people to participate in PE and be less inactive in China, which is consistent with those findings in other countries, such as Canada [32], Brazil [33], and Mexico [34]. In addition, the nudging effect of sports facilities construction on PE is negatively moderated by mental health status. Understanding the functions and limitations of the construction of sports facilities can guide governments’ policies and interventions for people’s inactivity.

Distinguished from the finding on the effect of diagnosed severe mental illness on PE participation in Denmark [52], the effect of general mental health is found to be insignificant on PE in China in this study. Mental health status may be neither a barrier nor motivator to people’s PE participation directly. Whether people feel depressed or not may have no significant relationship with their PE participation. However, mental health status could weaken or restrain the function of sports facility construction on individuals’ PE participation. People with poor mental health status may be inactive even surrounded by sports facilities.

Men and women do not participate equally in PE, as observed in previous study, and men are more attracted to the sports activities [60]. The gender-dependent variations may be attributed to the different motivations for PE of males and females in their childhood and cultural norms [61,62,63]. In Chinese culture, females are presupposed to be gentle and caring and they shoulder a higher share of housework load in families [64]. This mainstream view of gender division of characteristics may continue to exert influence on PE participation. In addition, PE participation is more prevalent among older people. Chinese elders are fond of doing PE to keep a healthy and active lifestyle, which is consistent with the finding in Western countries [65]. For instance, old people are found to engage in tai-chi sessions regularly for health in England [65].

The effect of educational level on PE participation is distinguished from existing literature, which finds that high-educated people are more likely to participate in physical activities compared with low-educated counterparts for individuals with better education may be more aware of the consequences of inactivity [66,67,68,69]. Nevertheless, an inversed pattern is found in China in this study that individuals with better education tend to be more inactive. It may be related to the design of the school education system in China, which pays much attention to students’ knowledge acquisition and little to PE. To obtain great grades, people with high educational level can hardly develop sports habit through this educational system. 

## 6. Limitations

This study has several limitations. The data used in this study are collected nationwide in 2014, so the generalization of the findings may miss the impact of major social events, such as the COVID-19 pandemic, which is hypothesized to have an impact on the mental health, physical activity, and sedentary behavior of people. In addition, policies for COVID-19-induced lockdown, worries about livelihood and being infected by disease, and social isolation are not considered in this study. Further research could be conducted to examine these factors on PE participation.

## 7. Conclusions

Firstly, this study provides the HPM with empirical evidence from China, which shows that both situational (sports facility construction) and personal (mental health status) factors can affect people’s participation in PE. The findings of thisstudy indicatethat the construction of sports facilities could significantly nudge people to participate in PE in China. Mental health status may have no direct significant effect on PE participation, while its moderating effect is observed in this study.

Secondly, one novelty of this study is the finding of mental health exerting an indirect effect on the relationship between construction of sports facilities and PE participation. Specifically, improved mental health can reinforce the nudging effect of built environment on PE. Therefore, it is insufficient for the government to encourage people to be active in sports only by building sports facilities. It is necessary for the government to comprehensively consider the macro factors (such as constructing sports facilities) and micro (such as mental health) factors and make an overall strategy for the development of PE. More considerations and psychological interventions should be given to people who are burned out by living pressure and stress caused by daily work and suffering from mental ill-health.

Thirdly, young, female, and better educated people composed the “neo-vulnerable” population, who are inactive in PE participation and need more interventions. Sports facilities should be improved to adapt to these “neo-vulnerable” populations’ need. For instance, more investment should be given to the design of female-friendly sports facilities. Dual career programs should be set up to attract and motivate more young and well-educated people to participate in various kinds of sports activities and cultivate stronger sports atmosphere in the future.

## Figures and Tables

**Table 1 healthcare-11-00219-t001:** Summary of the sample (*n* = 4634).

Variable	Mean	SD	Min.	Max.
Time of PE	7.621	8.996	0.1	98
Density of sports facilities	1.899	4.398	0	40
Gender	0.515	0.500	0	1
Age	1.797	0.688	1	3
Education	8.773	4.610	0	22
Marital status	2.029	0.850	1	5
Self-rated health status	2.916	1.198	1	5
BMI	22.990	3.492	8.642	45.369
Mental health status	4.550	0.607	1	5

**Table 2 healthcare-11-00219-t002:** Pearson correlation matrix of the study.

Variables	1	2	3	4	5	6	7	8	9
1. Time of PE	1								
2. Density of sports facilities	0.084 ***	1							
3. Gender	0.024	0.011	1						
4. Age	0.135 ***	0.040 **	0.003	1					
5. Education	−0.128 ***	0.110 ***	0.150 ***	−0.434 ***	1				
6. Marital status	0.055 ***	0.027	−0.095 ***	0.457 ***	−0.301 ***	1			
7. Self-rated health	0.038 **	0.067 ***	−0.081 ***	0.340 ***	−0.215 ***	0.186 ***	1		
8. BMI	0.015	0.005	0.069 ***	0.115 ***	−0.041 **	0.135 ***	0.064 ***	1	
9. Mental health status	−0.018	0.038 **	0.069 ***	−0.005	0.119 ***	−0.026	−0.241 ***	0.047 **	1

Notes: *** *p* < 0.001, ** *p* < 0.05; *n* = 4634.

**Table 3 healthcare-11-00219-t003:** Multiple linear regression results for PE time and density of sports facilities and mental health status.

Time of PE	Model 1	Model 2	Model 3	Model 4
Gender	0.639 **	0.648 **	0.656 **	0.679 **
	(0.269)	(0.268)	(0.268)	(0.268)
Age	1.365 ***	1.272 ***	1.300 ***	1.333 ***
	(0.246)	(0.250)	(0.251)	(0.248)
Education	−0.185 ***	−0.216 ***	−0.212 ***	−0.214 ***
	(0.036)	(0.038)	(0.037)	(0.037)
Marital status	−0.168	−0.197	−0.199	−0.216
	(0.184)	(0.182)	(0.182)	(0.181)
Self-rated health	−0.088	−0.138	−0.170	−0.177
	(0.125)	(0.122)	(0.122)	(0.122)
BMI	−0.002	−0.001	0.001	0.005
	(0.036)	(0.036)	(0.036)	(0.035)
Density of sports facilities		0.191 ***	0.192 ***	1.448 **
		(0.063)	(0.063)	(0.588)
Mental health status			−0.244	0.219
			(0.277)	(0.325)
Density of sports facilities * Mental health status				−0.275 **
				(0.123)
Constant	7.117 ***	7.362 ***	8.432 ***	6.262 ***
	(1.004)	(1.011)	(1.604)	(1.681)
R-squared F	0.026 20.93 ***	0.034 20.53 ***	0.034 18.20 ***	0.041 16.78 ***

Notes: *** *p* < 0.001, ** *p* < 0.05, * *p* < 0.1; *n* = 4634.

## Data Availability

The data analyzed in this study are available on reasonable request to the corresponding author.

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
