# Peer review of "Will the Construction of Sports Facilities Nudge People to Participate in Physical Exercises in China? The Moderating Role of Mental Health"

_healthcare, 2023, doi:10.3390/healthcare11020219_

Round 1

Reviewer 1 Report

I greatly appreciate the choice of topic and the effort put into this text by the authors.

I recommend that you write something about research among young people in the introduction, e.g. about academic sports, including dual careers.
It is worth referring more to research from other countries, especially from cultures other than China.
Do other countries have similar physical activity policies as China? Yes or no, it should be mentioned.
Could the COVID-19 pandemic have affected your involvement in the EP?

Great that there's a reference to a Cronbach
In methods, I would like to learn more about the questionnaire used in the study. What specific issues does it address? How were the subjects recruited? I know the data is from a public source, but maybe you have access to such information.
Do you know if people under the age of 18 were examined with the consent of their legal guardians? Relevant from the point of view of research ethics.
In the discussion, it is necessary to refer directly to the results, data from countries other than China.
Any other conclusions?
I miss the limitations of this study (e.g. no impact, including social, of the COVID-19 pandemic). You should write about them

Reviewer 2 Report

This was reviewed with interest. I have some important questions regarding the manuscript along with some comments to hopefully clarify some determining factors.

Abstract: It is recommended that the objective of the study be reformulated. It is also recommended to use the term, "the results of this study suggest", instead of "this paper”.

Introduction:

- Line 2: reference or rephrase the sentence.

- All the data provided by the plans established in China, it would be advisable to put them in context with proposals from other countries.

- It is recommended to reformulate the link between the concepts: sports practice and mental health.- In general, it is recommended to review more recent evidence for certain statements in this section.

- A better presentation of the hypothesis and objectives of the study is necessary. It is difficult for the reader to understand.

Literature review:

- There is a lack of connection between the concepts under study: promotion of the practice of physical activity through the construction of sports facilities and mental health.

- It is recommended to add references to support the statement “most studies have done in western and developed countries.”

Data and Methodology:

- More detail in the sample data is required

- A much greater level of detail is required when presenting the methodology for collecting results.

Results:

- How was the statistical power of this study calculated?

- What program was used to perform the analyses?

- Has the distance or distance radius of the effect of the facilities on citizens been defined?

 Discussion:

- Please begin this section by explaining whether or not the objectives of this study have been achieved.

- It is necessary to reinforce the statements on the studies of other countries with bibliographic references.

- The correlation between mental health and the construction of sports facilities requires further analysis and explanation, with greater support from scientific evidence.

Round 2

Reviewer 2 Report

Big improvement. Congrats